# A Prospective Multicenter Longitudinal Analysis of Suicidal Ideation among Long-COVID-19 Patients

**DOI:** 10.3390/healthcare12030290

**Published:** 2024-01-23

**Authors:** Alessandra Costanza, Andrea Amerio, Andrea Aguglia, Luca Magnani, Jacques Alexander, Alessandra Maiorano, Hélène Richard-Lepouriel, Elena Portacolone, Isabella Berardelli, Maurizio Pompili, Gianluca Serafini, Mario Amore, Khoa D. Nguyen

**Affiliations:** 1Department of Psychiatry, Faculty of Medicine, Geneva University (UNIGE), 1211 Geneva, Switzerland; helene.richard-lepouriel@hcuge.ch; 2Department of Psychiatry, Adult Psychiatry Service, Geneva University Hospital (HUG), 1205 Geneva, Switzerland; jacques.alexander@hcuge.ch (J.A.); alessandra.maiorano@hcuge.ch (A.M.); gianluca.serafini@unige.it (G.S.); mario.amore@unige.it (M.A.); 3Department of Neuroscience, Rehabilitation, Ophthalmology, Genetics, Maternal and Child Health, Section of Psychiatry, Faculty of Medicine, University of Genoa, 16132 Genoa, Italy; andrea.amerio@unige.it (A.A.); andrea.aguglia@unige.it (A.A.); 4IRCCS Polyclinic Hospital San Martino, 16132 Genoa, Italy; 5Department of Psychiatry, San Maurizio Hospital of Bolzano, 39100 Bolzano, Italy; magnani1991@gmail.com; 6Mood Disorder Unit, Department of Psychiatry, Psychiatric Specialties Service, Geneva University Hospital, 1205 Geneva, Switzerland; 7Institute for Health & Aging, University of California (UCSF), San Francisco, CA 94158, USA; elena.portacolone@ucsf.edu; 8Department of Neurosciences, Mental Health and Sensory Organs, Suicide Prevention Centre, Sant’Andrea Hospital, Faculty of Medicine and Psychology, Sapienza University, 00185 Rome, Italy; isabella.berardelli@uniroma1.it (I.B.); maurizio.pompili@uniroma1.it (M.P.); 9Department of Biomedical Sciences, Faculty of Life Sciences, Chinese University of Hong Kong, Hong Kong 999077, China; khoa.d.nguyen@gmail.com

**Keywords:** suicide, suicidal ideation, suicidal behavior, long COVID-19, neuroinflammation, chronic inflammation, prevention, mental health, prospective longitudinal analysis, multicentric study

## Abstract

Long coronavirus disease 19 (COVID-19) is an emerging multifaceted illness with the pathological hallmarks of chronic inflammation and neuropsychiatric symptoms. These pathologies have also been implicated in developing suicidal behaviors and suicidal ideation (SI). However, research addressing suicide risk in long COVID-19 is limited. In this prospective study, we aim to characterize SI development among long-COVID-19 patients and to determine the predictive power of inflammatory markers and long-COVID-19 symptoms—including those of psychiatric origin—for SI. During this prospective, longitudinal, multicenter study, healthy subjects and long-COVID-19 patients will be recruited from the University Hospital of Geneva, Switzerland, the University of Genova, the University of Rome “La Sapienza”, and the University of San Francisco. Study participants will undergo a series of clinic visits over a follow-up period of 1 year for SI assessment. Baseline and SI-onset levels of inflammatory mediators in plasma samples, along with 12 long-COVID-19 features (post-exertional malaise, fatigue, brain fog, dizziness, gastrointestinal disturbance, palpitations, changes in sexual desire/capacity, loss/change of smell/taste, thirst, chronic cough, chest pain, and abnormal movements) will be collected for SI risk analysis. The proposed enrollment period is from 15 January 2024 to 15 January 2026 with targeted recruitment of 100 participants for each study group. The anticipated findings of this study are expected to provide important insights into suicide risk among long-COVID-19 patients and determine whether inflammation and psychiatric comorbidities are involved in the development of SI in these subjects. This could pave the way to more effective evidence-based suicide prevention approaches to address this emerging public health concern.

## 1. Introduction

### 1.1. Etiology of Suicide

Suicide is a complex neuropsychiatric condition that ranks among the most prevalent causes of mortality and years lived with disability worldwide. While psychological and pharmacological support could address the psychiatric symptoms of individuals with suicidal ideation (SI) and suicidal behavior (SB), current suicide prevention approaches are of limited efficacy. This limitation is due to the absence of biomarkers for SI/SB with meaningful utility for clinically monitoring the progression of these conditions in individuals at risk [1,2]. Therefore, further elucidation of the multifaceted etiopathogenesis of SI/SB is urgently needed to develop a more effective SI/SB risk mitigation strategy.

There are multiple hypothetical etiologies of SI/SB, ranging from genetic/epigenetic abnormalities to neuroanatomical/biochemical alterations at virtually all physiological levels [3,4]. In this regard, clinical evidence suggests associations between increased suicide risk, immunological inflammation, and neurometabolic dysregulation. This has prompted further mechanistic elucidation of potential molecular and cellular contributors to these putative origins of SI/SB [5]. Furthermore, these neurobiological pathways often interact with each other and with other psychosocial determinants of SI/SB in individuals with pre-existing neuropsychiatric and somatic illnesses, leading to the development of these conditions [6,7,8,9].

### 1.2. Suicide Risk in Patients with Psychiatric and Physical Illnesses

Psychiatric disorders have been considered some of the most powerful predictors of SB/SI. In fact, among suicide victims (including both individuals with suicidal attempts and fatal suicides), more than 50% experienced major depression, while approximately 25–50% and 5–15% suffered from substance abuse and schizophrenia, respectively [9]. Furthermore, somatic illnesses that could predispose an individual to SI/SB development include neurological diseases, inflammatory bowel disease, diabetes, stroke, chronic obstructive pulmonary disease, osteoporosis, and chronic pain. These conditions have been identified as independent predictors of SI/SB [10,11,12]. Elevated suicide risk has been also associated with infections by various pathogens, including *Toxoplasma gondii* and Cytomegalovirus [13,14]. A potential real-time clinical insight that could enhance our understanding of the complex interplay between psychological and biological factors contributing to suicidality is the possible increased risk of SB/SI among patients experiencing coronavirus disease 2019 (COVID-19) [15,16]. This infectious illness is characterized by acute physical symptoms and enduring neuropsychiatric consequences [17,18,19].

### 1.3. Suicide Risk and COVID-19

COVID-19 originated in Wuhan, China, as a cluster of pneumonia-like illnesses in late 2019 and has since evolved into a global pandemic, affecting more than 700 million people include the reference. This lethal respiratory disease is caused by SARS-CoV2 infection, presenting acute clinical hallmarks of respiratory failure and multisystem inflammation. Furthermore, consistent with a previous observation of a neurotropic impact of vascular abnormalities in patients with neurological illnesses, SARS-CoV2-induced acute dysregulation of host immune response in the systemic vasculature could predict the presence of chronic symptoms of fatigue, pain, cognitive impairment, and other neuropsychiatric manifestations [20,21,22]. This condition has recently been termed long COVID-19. Importantly, it is hypothesized that the presence of overlapping neuroinflammatory features of long COVID-19 and SB/SI may lead to a potentially elevated suicide risk in this patient population.

During the early phases of the COVID-19 outbreak, the scientific community raised concerns about an elevated suicide risk [21,23,24,25]. This concern has been attributed to the convergence of various psychosocial factors, including economic crisis, social isolation, gender, age, quality of living environment, fear/anxiety of death, and various pre-existing psychiatric risks in concerned individuals [26,27,28,29]. For example, earlier case reports and observational studies have documented a rise in SI/SB during the first wave of COVID-19, mainly reported in emergency departments [30,31,32,33]. However, subsequent large-scale analyses have largely failed to confirm these initial findings [34,35,36,37]. This discrepancy might be explained by the absence of a direct and acute impact of SARS-CoV2 infection on vulnerable neurobiological circuits underlying SI/SB development. However, while a direct and acute impact may be absent, it is essential to consider that indirect effects, such as chronic residual low-grade inflammation, the use of certain medications like chloroquine, and the gradual development of psychiatric symptoms, including depression and anxiety [15,38,39,40,41,42,43], may still play a significant role in predicting a possible rise in SI/SB risk during the post-acute phase of SARS-CoV-2 infection.

Several inflammatory signatures associated with suicide risk have been observed in the pathogenesis of COVID-19. Notably, the cytokine storm resulting from hyperinflammation in COVID-19 patients, characterized by markedly elevated levels of pro-inflammatory cytokines such as IL1, IL6, and TNFa, has been linked to an increased risk of suicide. Activated monophagocytes, mast cells, and glial cells are identified as common crucial contributors to the hyperinflammatory syndrome observed in both subjects with an elevated suicidal risk and severe COVID-19 cases. Additionally, the widespread expression of ACE2, the cellular entry receptor for SARS-CoV2, suggests potential direct infection in various tissues, including the brain, leading to dysregulated peripheral tissue inflammation and neurological complications. These findings support the hypothesis that there is an intricate interplay between hyperinflammation, suicidal risk, and the pathophysiology of COVID-19 [15] and long-Covid [44,45,46] As a further confirmation, a cross-sectional study of American adults revealed significantly higher SI/SB scores in subjects with a previous clinical diagnosis of COVID-19 (*n* = 39) one year after infection compared to those without, despite a similar history of diagnosed depression and anxiety [47]. Similarly, in a retrospective analysis of electronic health records of 153,848 COVID-19 survivors in the US, SI risk one year after infection was 46% higher in the group that had been affected by SARS-CoV-2 compared to the control group of unaffected patients [21,48]. Collectively, these findings not only lend further clinical evidence supporting an elevated risk of SI/SB in the post-acute phase of COVID-19 but also underscore the need for a more in-depth characterization of the development of these neuropsychiatric symptoms in clinically well-defined long-COVID-19 patients [49].

### 1.4. Objective

This prospective case–control study aims to conduct one of the first longitudinal analyses of SI development among clinically well-defined long-COVID-19 patients, comparing them to healthy individuals. The primary objectives are to determine the prevalence and features of SI over a one-year follow-up period and to identify potential biological and neuropsychiatric predictors of SI onset in the context of long COVID-19. The study aims to contribute empirical evidence to the ongoing discourse on the mental health implications of COVID-19, with a specific focus on the persistent symptoms associated with long COVID-19 and their potential impact on suicide risk.

## 2. Materials and Methods

### 2.1. Ethics Approval

This study will undergo the regulatory approval process from the Institutional Review Board of the participating institutions. It will be conducted in accordance with the outlined research plan, complying with legal and regulatory requirements issued by the Public Health Department of Switzerland and respected institutions. Furthermore, the study will adhere to the current principles of the Declaration of Helsinki for human subject research. Prospective participants will receive consent forms outlining their rights and responsibilities as study participants and providing information on the overall objective and procedures of the research project. They will be given sufficient time to review the forms, and only those with written informed consent will be enrolled to participate in the study.

### 2.2. Study Participants

Study participants will be recruited at the Department of Psychiatry of the participating institutions. The study will include a cohort of long-COVID-19 patients and another cohort of healthy control subjects. Long-COVID-19 patients will include those with a history of SARS-CoV2 infection (confirmed by molecular diagnostic tests) and persistent symptom(s) of long COVID-19 at least 3 months after COVID-19 recovery. On the other hand, the healthy control group will include individuals with no prior SARS-CoV2 infection.

The inclusion criteria are outlined as follows: (1) subjects aged 18 years or older and not pregnant, (2) subjects without a history of SB/SI or current symptoms of SI, (3) subjects able to read and comprehend French/Italian/English depending on the recruitment sites, (4) subjects with a minimum weight of 100 lbs and willingness to donate 10 mL of venous blood twice (at baseline timepoint and SI onset), and (5) subjects willing to provide written informed consents. Prospective participants younger than 18 and unable to provide written informed consent or blood samples will be excluded from the study. In case of consent withdrawal or noncompliance, study subjects will be excluded, and additional participants will be recruited. Healthy control subjects who become infected with SARS-CoV2 or long-COVID-19 patients who cease to exhibit any long-COVID-19 symptoms at any assessment time points will also be excluded from the final analysis.

### 2.3. Study Design

Prospective participants will be given consent forms to be reviewed within 24 h. Subjects who agree to the conditions of the research project through written informed consent will be invited to a series of clinic visits. The study will involve administering a structured questionnaire to the participants aimed to collect socio-demographic information, such as age, gender, education level, marital status, occupation, and monthly income. To enhance the study’s accuracy, these demographic variables will be held constant as they are known to be associated with suicidal behaviors [50] This will allow for better isolation of the effects of long COVID-19 and its associated symptoms on SI development. Next, enrolled participants will undergo a confirmatory assessment for (1) the absence of both SI symptoms and/or physical/psychiatric illnesses, as the absence of SI and physical/psychiatric illnesses needs to be confirmed in both cohorts (see reviewer 2 comment 2) and (2) the absence of SI symptoms and the presence of long COVID-19 symptoms (for the long COVID-19 cohort). Participants who meet the confirmed eligibility criteria will proceed to provide a baseline blood sample. Following the initial visit, both long-COVID-19 patients and healthy controls will undergo quarterly follow-ups for one year to assess the potential development of SI. In the event of detecting the onset of SI, participants will be asked to provide a second blood sample to analyze changes in circulating cytokines compared to baseline. Participants who develop SI or exhibit high levels of psychological distress during the screening for psychiatric illnesses will receive tailored psychotherapeutic and pharmacologic support according to the standard of care of the Department of Psychiatry of the participating institutions. The overview of the study is depicted in the flowchart in Figure 1.

### 2.4. Anticipated Outcomes

The primary study outcomes involve the detection and comprehensive assessment of SI encompassing an in-depth evaluation of its features, including severity, throughout the one-year follow-up period. The secondary outcomes are the number and type of long COVID-19 symptoms and plasma levels of various inflammatory markers at the onset of SI. This multifaceted approach will enable a thorough understanding of the complex interplay between SI, long COVID-19 symptoms, and inflammatory markers, contributing valuable insights to the broader field of research.

### 2.5. Diagnosis of Long COVID-19

Long COVID-19 will be based on the presence of any of the twelve most frequent symptoms, recently identified and validated as clinical hallmarks of long COVID-19 [49]. These symptoms, comprising post-exertional malaise, fatigue, brain fog, dizziness, gastrointestinal disturbance, palpitations, changes in sexual desire/capacity, loss/change of smell/taste, thirst, chronic cough, chest pain, and abnormal movements, will be evaluated using self-assessment questionnaires completed by each subject during the clinic visit. The number and type of long COVID-19 symptoms will be recorded during each visit for downstream analysis.

### 2.6. Clinical Assessment of Psychiatric Illnesses

To confirm the absence of psychiatric illnesses, prospective participants from both cohort groups will undergo a confirmatory screening procedure at the first clinic visit. This procedure includes several questionnaires, namely the Insomnia Severity Index (threshold: 14 points) [51], the Hospital Anxiety and Depression Scale-Anxiety subscale (threshold: 7 points) [52], the Beck Depression Inventory-13 (threshold: 7 points) [53], and the post-traumatic stress disorder Check-List (threshold: 30 points) [54]. These questionnaires assess symptoms of insomnia, anxiety, depression, and post-traumatic distress, respectively. Prospective participants scoring below these thresholds—chosen to maximize the screening sensitivity for psychiatric symptoms [51,52,53,54]—will qualify for inclusion in either cohort group based on their SARS-CoV2 infection history.

### 2.7. Clinical Assessment of SI

To detect the presence of SI, participants will undergo screening using the Beck Depression Inventory-II (BDI-II) [55] questionnaire. The BDI-II consists of a 21-item multiple-choice test, each item corresponding to a particular symptom of depression. Participants are required to select one of four statements that most accurately reflects their emotions during the preceding two weeks, encompassing the day of the assessment. The responses are scored on a scale that spans from 0 to 3, with higher scores indicating more severe depressive symptoms. Individuals who provide a positive response to question 9 (pertaining to Suicidal Thoughts or Wishes) and score within the range of 1–3 on the other questions will undergo further clinical evaluation for the characterization and severity of SI. This evaluation will be conducted by a qualified psychiatrist employing the Scale for Suicidal Ideation (SSI) [56]. The SSI comprises a series of structured questions that aim to elicit information about the subject’s thoughts, plans, and attitudes regarding suicide. It is widely used to gain insight into the severity of SI, aiding in risk assessment and the development of appropriate intervention strategies. The scale’s structured format allows for standardized evaluation across different individuals, facilitating comparisons and analysis.

### 2.8. Blood Sample Collection

Peripheral blood (10 mL) will be obtained from participants through venipuncture and collected into di-potassium ethylenediaminetetraacetic acid (K2EDTA) tubes. The collected blood samples will be stored at room temperature until further processing for plasma sample collection as follows. Whole blood will be centrifuged at room temperature at 1800 rotations per minute (rpm), and the resulting plasma layer will be collected and stored in 2 to 3 aliquots of 2 mL Eppendorf tubes at −80 °C, each labeled with a unique identification code for each patient and date of collection of each patient and the collection date. Plasma samples will be then used for the analysis of soluble markers of inflammation.

### 2.9. Plasma Inflammatory Mediator Measurement

Various inflammatory cytokines implicated in the pathogenesis of COVID-19, including IL-1β (#558279), IL-6 (#558276), TNF-α (#558273), and IFN-γ (#558269) [25], will be analyzed from participants’ plasma samples by Cytometric Bead Array (BD Biosciences, assay range: 10–2500 pg/mL). Briefly, plasma samples will be incubated first with a mixture of fluorescence-coded capture beads and subsequently with detecting antibodies for specific cytokines. Unbound reagents will be washed away, and samples will be analyzed with a flow cytometer to measure the levels of circulating inflammatory cytokines.

Besides these cytokines, selected circulating markers of long-COVID-19-associated brain injury [57], including NFL, GFAP, MMP-9, and PPIA, will be assessed. Briefly, NFL (assay range: 0–2000 pg/mL) and GFAP (assay range: 0–40 ng/mL) will be quantified by the NEUROLOGY 2-PLEX B (#103520, Quanterix) on an SR-X Analyzer according to the manufacturer’s instructions MMP-9 (#DMP900, assay range: 0.3–20 ng/mL), and PPIA (#RD191329200R, assay range: 0.39–25 ng/mL) will be analyzed using ELISA kits from R&D Systems and Biovendor, respectively. Subsequently, quantification will be performed utilizing a colorimetric plate reader following standard protocols.

During the analysis of cytokines and markers of brain injury, if samples yield soluble protein readouts that exceed the detection limits of the assays, they will be appropriately diluted for further analysis.

### 2.10. Serious Adverse Events

Serious adverse events are defined as (1) any unfavorable occurrences for which a causal relationship with participants’ health data collection could not be excluded, (2) incidents requiring hospitalization/extension of hospitalization, (3) events causing persistent/significant disability or incapacity, (4) episodes of a life-threatening nature or those resulting in death. Might any of these adverse events occur, the project will be suspended and undergo further assessment. These events will be documented and reported to the ethics committee of the participating institutions within 7 days after their occurrence, for further evaluation of possible causal relationships with the research project. If the event has a plausible relationship to the research project and cannot be explained by underlying illnesses or other factors, it is considered “related” to the study. Conversely, if the event lacks specific connections to the project and can be explained by underlying illnesses or other factors, it is considered “unrelated” to the study. If applicable, further measures will be implemented to prevent the recurrence of such serious adverse events in future studies.

### 2.11. Data Management

Data generation, sharing, and storage strictly adhere to the regulations of the participating institutions involved in the study. Demographic and clinical data of each participant will be linked to a unique code and inserted manually into SPSS, version 23.0 (IBM), using double data entry to minimize errors. Patient information will be anonymized and stored with a numerical identification code corresponding to their medical records. Hard copies of patient-related documents will be securely stored with a physical lock for protection. Electronic data will be stored on an institutional server with strong password-encrypted protection. Patient privacy will be rigorously maintained, ensuring complete anonymity for any data presented at scientific forums or published in journals. Additionally, patient information will remain confidential and will not be disclosed to third parties during and after the study.

### 2.12. Data Analysis

The target size for both the healthy control and long COVID-19 cohorts is set at 100 each, with considerations for a 5% margin of error, a 95% confidence level, and an anticipated 10% dropout rate. Data will be analyzed with SPSS software, version 23.0 (IBM). Descriptive statistics including the central tendency (mean, median, mode), dispersion (range, variance, and standard deviation), and shape of the distribution of numerical datasets, as well as frequencies for categorical data, will be employed to provide a comprehensive summary of the demographic and clinical data of the study participants. Little’s MCAR (Missing Completely At Random) test will be performed to examine the presence of significant patterns in missing data. The multiple imputation functionality provided by SPSS will be used to generate imputed values for missing scores. The significance threshold for all statistical analyses will be set at two-tailed *p* < 0.05.

Regarding the primary outcome, the Z-test of proportions, an appropriate statistical approach that compares the means of the two independent samples to determine whether the observed difference is statistically significant, will be used to evaluate SI rates of healthy controls and patients with long COVID-19. Additionally, a 2 (group: long COVID-19, healthy control) × 4 (quarterly measurements) repeated-measures ANOVA will be used to monitor the evolution of potential SI over time in the two groups.

Once the presence or absence of SI in the respective groups is assessed, a chi-square (χ²) test for categorical data can be employed to assess whether there is a significant association between the presence of long COVID symptoms and the apparition of SI. The analysis of circulating inflammatory markers (at baseline and SI onset) requires a statistical approach that can accommodate the repeated measures nature of the data. In this regard, a 2 (time: baseline, onset) × 2 (group: COVID-19, control) repeated-measures ANOVA will be performed.

Multivariate linear and logistic regression models will be used to analyze the predictive power of plasma inflammatory markers and long COVID-19 symptoms for SI development. Additionally, a Cox proportional hazards regression will be used to test whether the baseline levels of inflammation would be predictive of the duration leading to SI onset.

## 3. Discussion

Suicidal ideation and behavior constitute a complex neuropsychiatric condition with profound global implications, ranking among the leading causes of mortality and years lived with disability. Despite advancements in psychological and pharmacological interventions, the limited efficacy of current suicide prevention approaches highlights the urgent need for a deeper understanding of the multifaceted etiopathogenesis of SI/SB. This imperative is underscored by the absence of reliable biomarkers for SI/SB, hindering effective clinical monitoring and mitigation strategies. Psychiatric disorders, including major depression, substance abuse, and schizophrenia, have long been identified as powerful predictors of SB/SI [9]. Furthermore, somatic illnesses such as neurological diseases, inflammatory bowel disease, diabetes, stroke, chronic obstructive pulmonary disease, and chronic pain are recognized as independent predictors of SI/SB [10,11,12]. Recent investigations have extended the spectrum to include infectious diseases like Toxoplasma gondii and Cytomegalovirus [13,14], emphasizing the broad range of factors contributing to suicide risk.

In the backdrop of this complex landscape, the global emergence of COVID-19 introduced a new layer of concern. As a pandemic affecting millions worldwide, COVID-19 presented not only acute respiratory symptoms but also chronic neuroinflammatory features previously associated with SI/SB. This prompted questions regarding the possibility of an increased suicide risk in this patient population.

Early reports indicated an increased suicide risk during the initial phases of the COVID-19 pandemic, attributed to various psychosocial factors such as economic crisis, social isolation, and fear of death [21,23,24,25,26,27,28]. However, subsequent large-scale analyses yielded conflicting results [34,35,36,37], suggesting a need for a more nuanced understanding of the relationship between COVID-19 and suicide risk. A recent 2023 study involving 6398 long-COVID-19 patients in the US provided the first comprehensive definition of long COVID-19. This definition relies on a set of 12 commonly (self)-reported symptoms, serving as the diagnostic foundation for this research project [49]. To date, only one cross-sectional study in Europe has established a connection between long COVID-19 and increased suicide risk. In this report, long-COVID-19 patients, characterized by a distinct set of symptoms (including fatigue, muscle weakness, pain, headache, respiratory and cognitive impairment, paresthesia, and anosmia) four months after hospitalization, were associated with increased risk of psychiatric disorders, new-onset psychiatric disorders, and SI development [58]. Overall, these findings prompt further exploration of the complex interplay between SI/SB and long COVID-19.

Furthermore, research has demonstrated a correlation between suicidal ideation and chronic physical illnesses. Chronic physical illness is linked to a diminished quality of life, giving rise to various practical, psychological, and social challenges. Depending on the specific condition, patients may experience pain, disability, disfigurement, and a demanding treatment regimen, with the repercussions extending to impairing one’s ability to engage in work or leisure activities. This, in turn, can lead to social isolation and heightened levels of anxiety and depression. For individuals living with their illnesses, an increase in suicidality emerges as a tangible concern. Existing evidence indicates an association between suicide and several medical conditions, including stroke, HIV/AIDS, stage 5 (end-stage) chronic kidney disease (stage 5 CKD), and multiple sclerosis (MS) [59,60,61,62,63,64,65,66,67,68]. In light of these findings, one can reasonably infer that long COVID-19, although not being explicitly classified as a chronic disease given its recent emergence, may be linked to an elevated SI risk.

Moreover, the COVID-19 pandemic has been a traumatic experience for many. Findings indicating an association between traumatic life events and suicidality [69], suggest a causal link between long COVID and suicidal ideation.

Enrollment for this study is scheduled to begin on 15 January 2024 and will run until 15 January 2026. The anticipated completion date for the study is 15 January 2027; however, potential delays may arise due to participant dropouts or exclusions of ineligible individuals during the follow-up period. With the overall one-year SI rate in Europe documented at 11% [70], it is expected that around 11 out of 100 individuals in the healthy control cohort may experience SI onset. This provides a comparative baseline, facilitating the characterization of SI development in long-COVID-19 patients.

Another important aspect of this research project involves conducting a mechanistic investigation to explore the possible influence of inflammatory pathology and neuropsychiatric comorbidities on SI development in long-COVID-19 patients. In this regard, specific proinflammatory cytokines, including IL-1β, IL-6, TNF-α, and IFN-γ, and markers of brain injuries, such as NFL and GFAP, have been proposed as etiological factors of SB/SI development [5,44,45]. Furthermore, while systemic increases in IL-1β, IL-6, TNF-α, and IFN-γ have traditionally been viewed as immunopathological hallmarks of acute COVID-19, most of these cytokines, along with several markers of brain injuries, have recently emerged as potential biochemical indicators of long COVID-19 [46,57]. Based on these observations, it is expected that the overlapping pathology of systemic and neuro-inflammation might potentiate SI development in long-COVID-19 patients [14], which is also accelerated by neuropsychiatric symptoms in these subjects. As such, our analysis of various markers of inflammation is expected to provide further insights on whether:specific alterations in inflammation markers can predict the development of SI;baseline levels of inflammation are correlated with the time it takes for SI to onset;SI-onset inflammation level is correlated with specific features/severity of SI.

Similarly, investigation of the predictive power of the number/type of long COVID-19 symptoms at baseline for SI development will be carried out. Additionally, correlation analysis to examine the relationship between features of long COVID-19 at SI onset and specific features/severity of SI will be carried out.

## 4. Conclusions

In conclusion, our prospective, longitudinal study on SI development among clinically well-defined long-COVID-19 patients is poised to make significant contributions to the existing literature. The emergence of long COVID-19 as a multifaceted illness with chronic inflammation and neuropsychiatric symptoms underscores the importance of investigating suicide risk in this patient population. By characterizing SI development and assessing the predictive power of inflammatory markers and long COVID-19 symptoms, our study aims to provide additional information to the existing literature on the potential causal relationship between long COVID-19 and SI.

The inclusion of healthy subjects and long-COVID-19 patients from multiple centers strengthens the generalizability of our findings. Through a rigorous assessment over a one-year follow-up period, we anticipate uncovering crucial insights into the intricate interplay of inflammation, psychiatric comorbidities, and SI development in long-COVID-19 patients.

The expected enrollment period from 15 January 2024 to 15 January 2026, with targeted recruitment of 100 participants for each study group, positions our study to offer robust and comprehensive data. The prospective nature of our analysis allows us to capture dynamic changes over time, providing a nuanced understanding of SI onset in long-COVID-19 patients.

The potential implications of our findings are significant. Confirmation of an increased SI risk in long-COVID-19 patients could serve as a significant driver for further research, leading to the development of more effective evidence-based suicide prevention approaches tailored to this specific patient population. In the broader context, our study will contribute to the evolving understanding of the long-term consequences of COVID-19, highlighting the importance of addressing not only the acute physical symptoms but also the potential neuropsychiatric sequelae associated with the illness.

## Figures and Tables

**Figure 1 healthcare-12-00290-f001:**
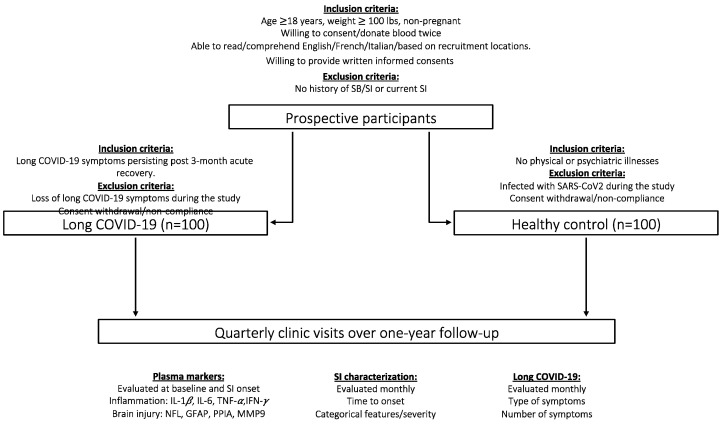
Overview of the protocol study.

## Data Availability

Not applicable.

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
