# Peer review of "A Prospective Multicenter Longitudinal Analysis of Suicidal Ideation among Long-COVID-19 Patients"

_healthcare, 2024, doi:10.3390/healthcare12030290_

Round 1

Reviewer 1 Report

Comments and Suggestions for Authors

Author,

The suggested study on "prospective multicenter longitudinal analysis of suicidal ideation among long COVID-19 patients" is a highly significant and much-needed investigation in the field of suicidology. The study's findings will undoubtedly enhance the existing body of literature on bio markers and SI. The present document provides a comprehensive description of the proposed project, which is scheduled to commence in the year 2024. There are some other elements that, if included, could offer further insight into the proposed manuscript. 1.The existing literature on this topic is limited. However, the author can contribute by proposing a mechanism that could perhaps explain the connection between these biomarkers and an elevated risk of suicide.

2. The sample has defined inclusion and exclusion criteria, indicating that the function of demographics has previously been deemed significant. It is important to clarify which demographic the author suggests examining and which demographic should be held constant in order to improve forecast accuracy.

The heading for the result section can be labelled as "anticipated outcomes".

4. The discussion part may encompass the validation of the suggested theoretical framework, which can be incorporated into the rationale or background.

May you encounter much success in this endeavour!

Author Response

Dear Reviewers,

Thank you for your thorough review of our manuscript titled "A prospective multicenter longitudinal analysis of suicidal ideation among long COVID-19 patients”. We appreciate the time and effort you invested in providing constructive feedback. Below, we address each of your comments:

Addressing the Comments to Reviewer 1

Comment 1: Propose a mechanism that could perhaps explain the connection between these biomarkers and an elevated risk of suicide.

Response: Thank you for your insightful comment. We have incorporated your suggestion and proposed a mechanism that could explain the connection between the observed biomarkers and an elevated risk of suicide. The added paragraph emphasizes the inflammatory signatures associated with suicide risk in the context of COVID-19 pathogenesis, linking hyperinflammation and the cytokine storm to an increased risk of suicide. The involvement of various cell types and the potential direct impact on the brain through ACE2 expression are discussed.

Comment 2: Clarify the demographic to be examined and specify which variables are to be kept constant to enhance forecast accuracy.

Response: Thank you for your suggestion, we have included the information you requested. The study will entail administering a structured questionnaire to participants to gather socio-demographic data, including age, gender, education level, marital status, occupation, and monthly income. To bolster the study's accuracy, these demographic variables will be held constant, as they have been identified in previous research as being associated with suicidal behaviors.

Comment 3: The heading for the result section can be labelled as "anticipated outcomes”.

Answer to the comment: We have amended this heading. We agree that this adjustment aligns with the prospective nature of the study, contributing to a more accurate representation of the research framework.

Comment 4: In the discussion include the validation of the suggested theoretical framework, which can be incorporated into the rationale or background.

Response: Thank you for this valuable feedback. We have incorporated an expanded discussion that delves into the validation of the suggested theoretical framework. The included section emphasizes the intricate nature of SI/SB, noting their global impact as leading causes of mortality and disability. It underscores the urgent need for a deeper understanding of the multifaceted etiopathogenesis of SI/SB, given the limited efficacy of current prevention approaches and the absence of reliable biomarkers for SI/SB. The discussion further explores the well-established predictors of SB/SI, ranging from psychiatric disorders to somatic illnesses, and extends the spectrum to include infectious diseases. Within this complex landscape, the emergence of COVID-19 globally introduces new concerns, as it not only presents acute respiratory symptoms but also chronic neuroinflammatory features previously linked to SI/SB.

Reviewer 2 Report

Comments and Suggestions for Authors

Author Response

Dear Reviewers,

Thank you for your thorough review of our manuscript titled "A prospective multicenter longitudinal analysis of suicidal ideation among long COVID-19 patients”. We appreciate the time and effort you invested in providing constructive feedback. Below, we address each of your comments:

Addressing the Comments to Reviewer 2

Comment 1: In the introduction, explicitly identify the symptoms of long COVID associated with SI/SB, providing a stronger causal justification for hypothesizing a relationship between long COVID and SI/SB.

Response: Thank you for this insightful comment. The revised manuscript now includes a discussion of inflammatory signatures in COVID-19 pathogenesis, emphasizing the intricate interplay between hyperinflammation, suicidal risk, and COVID-19's pathophysiology. Moreover, indirect effects, such as chronic inflammation and medication use, contributing to a potential rise in suicidal ideation/behavior during the post-acute phase of SARS-CoV-2 infection, are discussed.

Comment 2: Section 2.2 regarding the inclusion criteria for the control and experimental groups is not clear. The first paragraph mentions that the control group should have "no physical/psychiatric illness," while the second paragraph introduces an inclusion criterion stating that subjects should have no current symptoms of SI, without specifying that this applies only to the control group. Clarify that the criterion applies to both groups.

Response: Thank you for pointing this out. Indeed, the section was not clear, we have amended the manuscript so now it is clearly stated that both cohorts will consist of participants without a history and/or current symptoms of SI. This inclusion criteria will help define the parameters of SI relation to long-COVID marking a starting point for future measuring the prevalence of SI in long-COVID patients.

Comment 3: On the section “Serious adverse events” replace the phrase "put on hold" with “suspended”.

Response: Thank you for mentioning this. We agree that the term "suspended" fits better in the context, and we have made the appropriate amendment.

Comment 4: The choice of statistical analyses may be inconsistent with the nature of the dependent variable (indications of SI/SB). Use an independent-samples t-test for continuous data rather than chi-square tests intended for categorical data.

Response: Thank you or this constructive comment. We agree to remove the chi-square (χ²) test, thank you for pointing it out. If the patients are placed in the "long COVID" group based on the presence or absence of symptoms, then it is redundant to perform a chi-square (χ²) test for the secondary outcome of long COVID symptoms. The chi-square test is typically used to investigate whether two categorical variables are independent or if there is an association between them. In our case, we are dealing with a single categorical variable ("long COVID symptoms: present or absent”), which has been already established. However, we believe it would be of relevance to use the chi-square test differently and here is how: once assessed the presence or absence of SI in the respective groups, a chi-square test can be employed to assess whether there is a significant association between the presence of long COVID symptoms and apparition of SI.

Reviewer 3 Report

Comments and Suggestions for Authors

Contributors to thoughts of self harm or suicidal ideation (SI) as reported by the BDI (Beck) scale are a snapshot or in other words they are reported for a brief time. This reporting is non-specific and authors may want to describe the scale called Scale for Suicidal in length. This will add to the discusion. Chronic illness overall can contribute to SI and explaining the novelty of COVID-19 in relation to other similar illnesses can add to the strength of the paper.  Furthermore, trauma as a variable to SI is a significant variable (https://www.ncbi.nlm.nih.gov/pmc/articles/PMC6136384/). 

Author Response

Dear Reviewers,

Thank you for your thorough review of our manuscript titled "A prospective multicenter longitudinal analysis of suicidal ideation among long COVID-19 patients”. We appreciate the time and effort you invested in providing constructive feedback. Below, we address each of your comments:

Addressing the Comments to Reviewer 3

Comment 1: Describe the scale called Scale for Suicidal in length.

Response: Thank you for your comment. We have added a thorough description of the Scale for Suicidal to ensure a comprehensive understanding of its dimensions and relevance within the context of the study.

Comment 2:  As chronic illness can contribute to SI, emphasize how COVID-19 can contribute to SI in this context.

Response: Thank you for your insightful comment; we find it particularly interesting. We have incorporated this valuable information into the article. We found that research consistently highlights the correlation between SI and chronic physical illnesses, attributing it to the various challenges such as diminished quality of life, pain, disability, and social isolation. This association extends to specific medical conditions like stroke, HIV/AIDS, stage 5 chronic kidney disease, and multiple sclerosis. Considering these parallels, it is reasonable to infer that long-COVID, while not explicitly classified as a chronic disease, may be associated with an elevated risk of suicidal ideation.

Comment 3: Consider trauma as a significant variable to SI risk.

Response: Thank you for this additional valuable comment. We have incorporated this information in the manuscript. Indeed, the COVID-19 has been a profoundly traumatic experience for many, and research highlighting an association between traumatic life events and suicidality implies a potential causal link between long COVID and the emergence of SI.

Reviewer 4 Report

Comments and Suggestions for Authors

When I started reading the paper I found it brilliant and exciting. The research is very interesting and I think it will be a great contribution to the scientific community. The objectives are clear and well defined. The methodology is very clear and well detailed. Up to this point, everything was fine. However, when I get to section 3 I see that there are no results. I found this puzzling and started to look through the paper again. I had not realised that it is a paper about a future research protocol or ongoing. In the second paragraph of the discussion it is clearly shown. In the second paragraph of the discussion this is clearly shown but not before. Even the way the objectives are explained does not hint at this. This makes me question the confusing way in which the article has been done. That is the main problem. I repeat, the research is thought-provoking, brilliant and very well written, but it looks like research that has been done. Instead, it is an article that hypothesises future results. For this reason, I believe that the authors should reconstruct this paper in this sense, giving it a different focus and incorporating much more information on previous work. The conclusions should also be expanded.

On the other hand, I would point out two minor points: 

- Line 75: Species names are written in italics, so Toxoplasma gondii should be in italics.

L- ine 216: Where it says "florescence-coded" it should read "fluorescence-coded".

Author Response

Dear Reviewers,

Thank you for your thorough review of our manuscript titled "A prospective multicenter longitudinal analysis of suicidal ideation among long COVID-19 patients”. We appreciate the time and effort you invested in providing constructive feedback. Below, we address each of your comments:

Addressing the Comments to Reviewer 4

Comment 1: Clearly indicate from the outset that the study is prospective, provide additional details on relevant prior research and expand the conclusion.

Response: Thank you for your review and insightful comments. We acknowledge the confusion regarding the paper's focus on a future research protocol. We have addressed this by revising the introduction and objectives sections to clearly convey the speculative nature of the study. Additionally, we have provided more contextual information from previous work in the discussion and expanded the conclusions for a more comprehensive summary.

Comment 2: Revise these two minor points: (i) on Line 75, italicize “Toxoplasma gondii”, and (ii) in Line 216, correct the typo in 'florescence-coded' to ‘fluorescence-coded.

Response: Thank you for your feedback. We made the requested revisions.

We believe that these revisions have strengthened the manuscript, and we are confident that the paper is now better positioned for publication. We remain open to further suggestions and are committed to making any additional changes deemed necessary.

Thank you once again for your valuable feedback.

Sincerely,

Round 2

Reviewer 4 Report

Comments and Suggestions for Authors

Before going into the second revision of the article, I would like to make a brief comment. I suggest to the authors that, the next time they have to make changes in their article, they do it with change control as it helps the revision. I have been able to read all the comments but found it somewhat difficult.

Let us now turn to the review of the amended text. The authors have made a great effort to improve the clarity of the article. Changes have been introduced to help the reader understand the research that has been done. Expressions have also been improved to avoid confusion.

Now there is a logical argument throughout the article. It is well understood, from the beginning, that it is a design study and not research conducted. Congratulations to the authors for their efforts.

The conclusion has been completely modified in line with the changes made and, again, is clearer and more appropriate following these changes.

I believe that the article can be published without any problem and I repeat my thanks and congratulations to the authors for the work done.